

# *Melanoleuca subgriseoflava* and *M. substridula*—two new *Melanoleuca* species (Agaricales, Basidiomycota) described from China

Yue Qi[1], Cai-Hong Li[1], Yu-Meng Song[1], Ming Zhang[2], Hong-Bo Guo[3] and Xiao-Dan Yu[1]

[1] College of Biological Science and Technology, Shenyang Agricultural University, Shenyang, Liaoning, China
[2] Guangdong Academy of Sciences, Guangdong Institute of Microbiology, Guangzhou, Guangdong, China
[3] College of Life Engineering, Shenyang Institute of Technology, Fushun, Liaoning, China

## ABSTRACT

Two new *Melanoleuca* species, *Melanoleuca subgriseoflava* and *M. substridula*, are originally reported and described in China based on both morphological and molecular methods. *Melanoleuca subgriseoflava*, collected in Liaoning province, is mainly characterized by its greyish-brown to yellowish-grey pileus, creamy to light orange lamellae, greyish-yellow context, round and warted basidiospores and fusiform hymenial cystidia. *Melanoleuca substridula*, discovered in Sichuan province, is mainly characterized by its light brown to dark brown pileus, whitish lamellae, light brown to greyish-brown stipe, round and warted basidiospores and lack of any forms of cystidia. The phylogenetic relationships as well as divergence-time estimation were analyzed using the combined data set (ITS-nrLSU-RPB2), and the results showed that the two *Melanoleuca* species formed two distinct lineages. Based on the combination of morphological and molecular data, *M. subgriseoflava* and *M. substridula* are confirmed as two new species to science. A theoretical basis is provided for the species diversity of *Melanoleuca*.

# INTRODUCTION

*Melanoleuca* Pat. is distributed worldwide, containing around 423 validly published names (Index Fungorum, http://www.indexfungorum.org/, accessed on 7 April 2022), 12 of which were considered as edible by *Dai et al. (2010)*, including *Melanoleuca arcuata* (Bull.) Singer, *Melanoleuca brevipes* (Bull.) Pat., *Melanoleuca cognata* (Fr.) Konrad & Maubl etc. Recently, many new species of *Melanoleuca* have been reported throughout the world (*Vizzini et al., 2010*; *Vizzini et al., 2011*; *Sánchez-García, Cifuentes-Blanco & Matheny, 2013*; *Antonín et al., 2014*; *Antonín et al., 2017*; *Antonín et al., 2021*; *Yu et al., 2014*; *Nawaz, Jabeen & Khalid, 2017*; *Xu et al., 2019*; *Pei et al., 2021*). *Melanoleuca* is a taxonomically complicated genus because many species in the genus are very similar in macroscopical characteristics and only present subtle differences (*Bon, 1991*; *Boekhout, 1999*; *Vizzini et al., 2011*). The genus is typified by a dull-colored pileus; amyloid and warted basidiospores; two types of cystidia

Corresponding author
Xiao-Dan Yu, yuxd126@126.com

(urticiform or fusiform to lageniform) and all hyphae without clamp connections (*Singer, 1986*; *Bon, 1991*; *Boekhout, 1988*; *Vizzini et al., 2011*).

Despite considerable evidence that the *Melanoleuca* genus belongs to a monophyletic group, the infrageneric classification system of the genus has always been controversial. *Singer (1986)* divided the genus into four sections circumscribed only by pileus color and stipe ornamentations, *i.e.*, sect. *Alboflavidae* Singer, sect. *Humiles* Singer, sect. *Oreinae* Singer and sect. *Melanoleuca* Pat. *Bon (1978)*, the first to take micro-morphological characters into consideration, divided the genus into seven sections. As *Boekhout (1988)* emphasized the crucial role of cystidia, the genus was divided into three subgenera according to the absence/presence and shape of cystidia, *i.e.*, subgen. *Macrocystis* Boekhout, subgen. *Melanoleuca* Pat. and subgen. *Urticocystis* Boekhout. However, these taxonomical units are not supported by molecular data. *Vizzini et al. (2011)*, using a large number of ITS sequences to construct phylogenetic relationships of *Melanoleuca*, defined only two subgenera, *i.e.*, subgen. *Urticocystis* Vizzini and subgen. *Melanoleuca* Vizzini. Then, follow-up studies on species of *Melanoleuca* support the classification opinion proposed by *Vizzini et al. (2011)* (*Yu et al., 2014*; *Kalmer, Acar & Dizkirici, 2018*; *Xu et al., 2019*).

Only 31 species of *Melanoleuca* have been reported in China (*Bau & Li, 1999*; *Zhang, Li & Song, 2001*; *Chen, 2007*; *Mao, 2009*; *Sun et al., 2012*; *Wang, 2013*; *He et al., 2014*; *Yu et al., 2014*; *Zhao et al., 2014*; *Wei, Fan & Yan, 2015*; *Du et al., 2016*; *Tian et al., 2018*; *Xu et al., 2019*; *Pei et al., 2021*). Although China has a complex climate and geographical conditions, species resources of *Melanoleuca* remain scarce. This study reports and describes two *Melanoleuca* species collected from Liaoning province and Sichuan province in China from 2019 to 2020. In order to confirm the two collections as new to science, both morphological and method analyses were conducted. The morphological similarities and differences between the two species and other related species are also discussed.

## MATERIAL AND METHODS

### Specimens and morphological description

Fresh basidiomata were photographed in the field. Specimens were dried with an electric drier and deposited with silica gel. Dried specimens were preserved in the Fungal Herbarium of Shenyang Agricultural University (SYAU-FUNGI), Liaoning, China and Herbarium of Microbiology Institute of Guangdong (GDGM), Guangdong, China. Tissue blocks were removed from the inner part of the dried specimens for DNA analyses. Color abbreviations followed *Kornerup & Wanscher (1963)*. Methods for morphological observation followed *Pei et al. (2021)*. For observation of surface of the basidiospores, SEM microphotographs were performed using a scanning electron microscope (REGLUS 8100; Hitachi, Tokyo, Japan).

### Nomenclature

The electronic version of this article in Portable Document Format (PDF) will represent a published work according to the International Code of Nomenclature for algae, fungi, and plants, and hence the new names contained in the electronic version are effectively published under that Code from the electronic edition alone. In addition,

**Table 1  Primers used in this study.**

| Regions | Primer | Sequence (5′–3′) | Reference |
|---------|--------|------------------|-----------|
| ITS | ITS5 | GGAAGTAAAAGTCGTAACAAGG | *White et al. (1990)* |
|  | ITS4 | TCCTCCGCTTATTGATATGC | *White et al. (1990)* |
| nLSU | LROR | GTACCCGCTGAACTTAAGC | *Michot, Hassouna & Bachellerie (1984)* |
|  | LR5 | ATCCTGAGGGAAACTTC | *Michot, Hassouna & Bachellerie (1984)* |
| RPB2 | b7.1R | TGGGGYATGGTNTGYCCYGC' | *Matheny et al. (2007)* |
|  | b6F | CCCATRGCYTGYTTMCCCATDGC | *Matheny et al. (2007)* |

new names contained in this work have been submitted to MycoBank from where they will be made available to the Global Names Index. The unique MycoBank number can be resolved and the associated information viewed through any standard web browser by appending the MycoBank number contained in this publication to the prefix "http://www.mycobank.org/MycoTaxo.aspx?Link=T{&}Rec=". The online version of this work is archived and available from the following digital repositories: PeerJ, PubMed Central SCIE, and CLOCKSS."

## Phylogenetic construction

Genomic DNA was extracted from the dried specimens using the CTAB method (*Doyle & Doyle, 1987*). PCR protocol and sequencing were conducted as described by *Wang et al. (2019)*. Primer pairs used in this study are listed in Table 1. The newly generated sequences were submitted to GenBank.

Representative sequences of *Melanoleuca* species in former studies (*Sánchez-García, Cifuentes-Blanco & Matheny, 2013*; *Yu et al., 2014*; *Antonín et al., 2014*; *Antonín et al., 2015*; *Antonín et al., 2017*; *Nawaz, Jabeen & Khalid, 2017*; *Xu et al., 2019*; *Antonín et al., 2021*; *Pei et al., 2021*) were retrieved from GenBank. These sequences were aligned with those obtained from this study using Bioedit v7.0.9 (*Hall, 1999*) and MAFFT v7.313 (*Katoh & Standley, 2013*). The data partition homogeneity test (*Farris et al., 1995*) performed in PAUP (*Swofford, 2003*) allowed combining three regions (ITS, nrLSU and RPB2) (*P*-value 0.43). A combined data set was then completed with *Pluteus romellii* as an outgroup. The data matrix includes a total of 2,073 characters of 71 samples. Bayesian Inference (BI) and Maximum Likelihood (ML) were performed as previously described in *Pei et al. (2021)*. Specifically, the combined data set was run for 2 million generations under the GTR+I+G mode using MrBayes v.3.2.6 (*Ronquist et al., 2012*). RAxML−8.2.10-WIN was performed under the GTR-GAMMA model of evolution (*Stamatakis, 2014*). The resulting files were viewed using Figtree v1.4.4 (*Rambaut, 2018*) and were compiled in Adobe Illustrator CC.

## Divergence time estimation within *Melanoleuca*

Divergence time was estimated using BEAST v2.6.3 (*Bouckaert et al., 2014*). BEAUTI v2.6.3 was used to construct an XML file. ModelFinder (*Kalyaanamoorthy et al., 2017*) was used to infer the best substitution model. The clock model and substitution model were chosen following *Pei (2021)* and *Zhao et al. (2016)*. On the calibrated nodes, the offset ages of 98 and 110 Ma were set for the genus *Melanoleuca* and *Pluteus*, respectively (*Pei, 2021*). We

**Table 2  The sequenced *Melanoleuca* species analyzed in this study.**

| Species | Voucher collection | Origin | GenBank accession No. | | |
|---|---|---|---|---|---|
| | | | ITS | nLSU | RPB2 |
| *M. subgriseoflava* | SYAU-FUNGI-073 | Shenyang City, Liaoning Province, China | ON262573 | ON262569 | ON220896 |
| *M. substridula* | GDGM 84648 | Jiuzhaigou valley, Sichuan Province, China | ON262575 | ON262571 | ON220898 |
| *M. subgriseoflava* | SYAU-FUNGI-074 | Shenyang City, Liaoning Province, China | ON262574 | ON262570 | ON220897 |
| *M. substridula* | GDGM 84683 | Jiuzhaigou valley, Sichuan Province, China | ON262576 | ON262572 | ON220899 |

ran four independent Monte Carlo Markov Chains (MCMC) of 50 million generations, logging states every 5,000 generations. The checking for convergence was completed in Tracer v1.6 (*Rambaut, 2018*). TreeAnnotator v.1.8 was used for summarizing tree files. The resulting files were viewed and compiled in Figtree v1.4.4 (*Rambaut, 2018*) and Adobe Illustrator CC, respectively.

## RESULTS

### Phylogenetic analyses

The GenBank accession numbers of the sequences, determined in this study, are from ON262569 to ON262573 and ON220896 to ON220899 (Table 2). Maximum likelihood (ML) and Bayesian Inference (BI) showed almost identical topologies and the BI tree was selected for display (Fig. 1). The phylogenetic result suggested that the *Melanoleuca* should belong to a monophyletic group (PP = 1, BS = 100), which is consistent with the previous studies (*Yu et al., 2014*; *Vizzini et al., 2011*). A total of five clades (Clade A to Clade E) can be recognized within *Melanoleuca*, which is in line with the result of *Pei et al. (2021)*. Additionally, the collections (SYAU-FUNGI-073 and SYAU-FUNGI-074) named *M. subgriseoflava* formed an independent lineage with strong statistical support (PP = 1.00, BS = 100), located within clade E. And these specimens are closely related to a clade containing sequences of http://www.indexfungorum.org/names/Names.asp?strGenus=Melanoleuca griseoflava X.D. Yu & H.B. Guo, *M. arcuata* (Bull.) Singer and *M. heterocystidiosa* (Beller & Bon) Bon. In clade A, the collections (GDGM 84648 and GDGM 84683) named *M. substridula* group together with well support (PP = 1.00, BS = 100) and far away from the other species in *Melanoleuca*.

Maximum Clade Credibility (MCC) tree for *Melanoleuca* (Fig. 2) generated a topology similar to those of the phylogenetic analyses. Two new species of *Melanoleuca* also formed two separate clades with high support (PP = 1.00).

### Taxonomy

***Melanoleuca subgriseoflava* X.D. Yu & H.B. Guo, sp. nov.**

MycoBank No. MB843803 (Fig. 3)

**Etymology** The epithet "*subgriseoflava*" refers to the greyish-brown color of the pileus, which is similar to the species *Melanoleuca griseoflava*.

**Diagnosis:** Pileus convex to applanate to depressed at center, greyish-brown to yellowish-grey pileus; lamellae adnate to sinuate, creamy to light orange; stipe yellowish-brown to

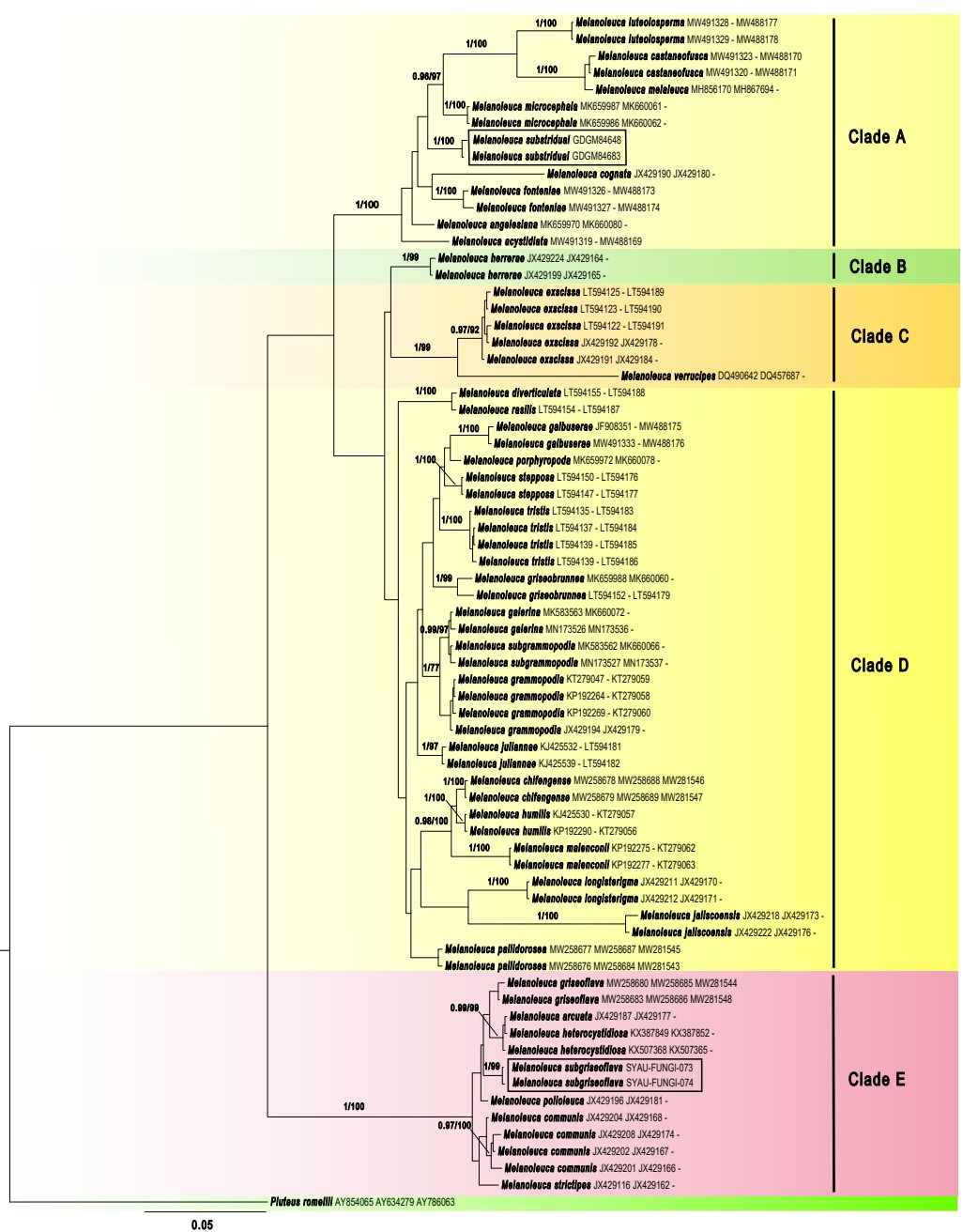

**Figure 1** Phylogenetic positions of the two new *Melanoleuca* species, inferred from the combined regions (ITS-nrLSU-RPB2) using MrBayes. The lineages with new species were shown in boxes. PP ≥ 0.95 and BS ≥ 75% were indicated around the branches. Accession numbers in GenBank (ITS, nrLSU, RPB2) follow the fungal names.

brown; context greyish-yellow; basidiospores with round and scattered warts and hymenial cystidia fusiform.

**Holotype:** CHINA. Liaoning Province: Shenyang City, Shenyang Agricultural University, on the soil in meadows, 2 Sep 2020, X.D. Yu (holotype: SYAU-FUNGI-073).

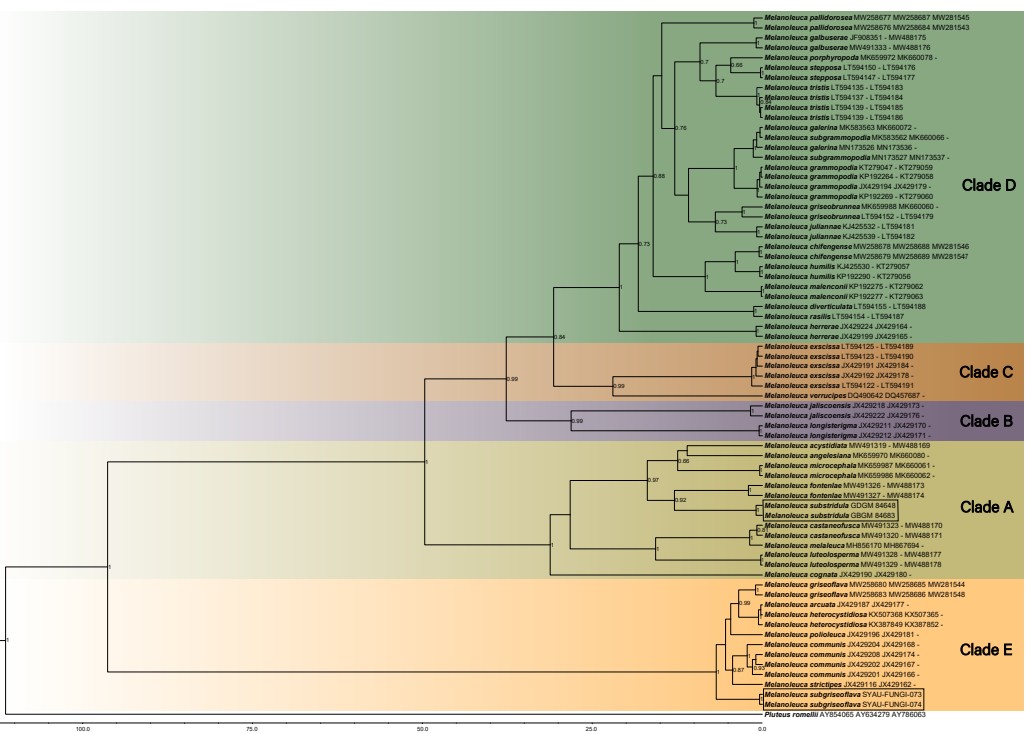

**Figure 2** Maximium Clade Credibility tree of *Melanoleuca* based on ITS, nrLSU, and RPB2 genes sequences with the outgroup *Pluteus.* The lineages with new species were shown in boxes. PP ≥ 0.60 are annotated at the internodes.

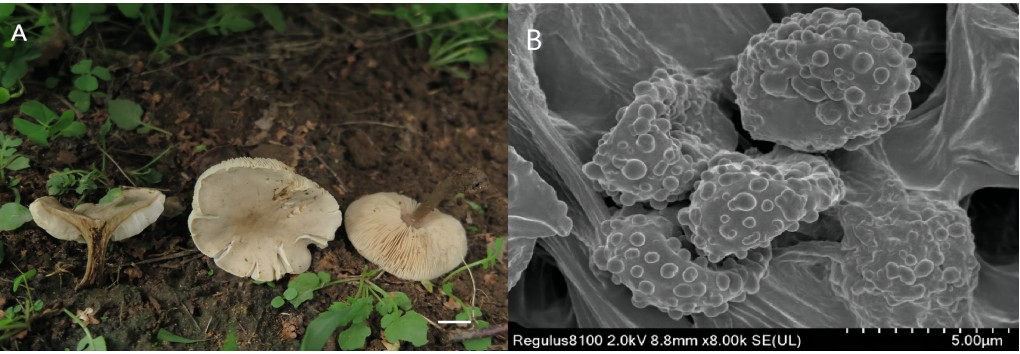

**Figure 3** *Melanoleuca subgriseoflava* (Holotype, SYAU-FUNGI-073). (A) macroscopic habitat (B) surface of basidiospores. Scale bars: 1 cm (A); 5 μm (B).

**Description:** Basidiomata medium-sized. Pileus 24–70 mm diam., convex to plano-convex at first, then gradually applanate, becoming depressed at center when mature; margin entire at first, and slightly lacerate when mature; surface greyish-brown (6D3 to 6F3) at first, then gradually becoming greyish-orange, greyish-yellow, yellowish-grey (5B3 to 5B5, 4B2 to 4B5, 4B2) when mature and dry, brownish-orange at centre (5C5 to 5C6). Lamellae adnate to sinuate, 3.0–7.0 mm broad, white to greyish-orange (5A1, 5B3) at first,

becoming creamy, orange-white, light orange (4A3, 5A2, 5A3) when mature, often deeper at margin, crowded, with lamellulae of two or four lengths, edge entire. Stipe cylindrical and somewhat broadened downwards, 25–55 mm long × 2.0-−7.0 mm diam., central, solid, light brown to yellowish-brown (5D7 to 5D8) in upper part, often becoming brown (5F8 to 6E8) towards base, with whitish flocculose apex, striate. Pileus context up to 10 mm thick near stipe attachment, thinner at margin, greyish-yellow (4B3 to 4B6), unchanging when exposed. Smell slightly farinaceous, taste mild. Spore deposit creamy.

Basidiospores (88/10/8) (5.0) 7.0–8.0 (8.5) × 4.0-−5.0 (6.0) μm, av. 7.6 × 4.9 μm, $Q = 1.55$, broadly ellipsoid, some obovate, ornamentation of small to large warts, some warts with irregular ridges, amyloid. Basidia (40/10/8) (18) 23–27 (28) × (7.0) 8.0–10.0 (11.0) μm, av. 26 × 8.7 μm, clavate to subclavate, hyaline, four-spored. Cheilocystidia lageniform, fusiform to conical cystidia, (30/10/8) (40) 45–49 (50) × (9.0) 10.0–11.0 (12.0) μm, found both at the edge of lamellae, most thin-walled, less thick-walled without distinct upper part. Pleurocystidia have a small amount, similar to cheilocystidia. Trama hyphae thin-walled, regular, 4.0–10.0 μm wide, inamyloid. Pileipellis hyphae cylindrical, with numerous branched, thin-walled, up to 10.5 μm wide. Stipitipellis hyphae in parallel, 4–14.0 μm wide, thin-walled, somewhat slightly thick-walled. Caulocystidia absent. Clamp connections absent in all tissues.

**Habitat and distribution:** Solitary or in small groups, saprotrophic on the soil, on the grass, on roadsides, in woods. Known from north-eastern China.

**Additional material studied:** CHINA. Liaoning Province: Shenyang City, Dongling Park, on the soil in meadows, 21 Jul 2019, H.B. Guo (SYAU-FUNGI-074); CHINA. Liaoning Province: Shenyang City, Dongling Park, on the grass in woods, 21 Jul 2019, X.D. Yu (SYAU-FUNGI-075); CHINA. Liaoning Province: Shenyang City, on the campus of Shenyang Agricultural University, on roadsides, 2 Sep 2020, X.D. Yu (SYAU-FUNGI-076); CHINA. Liaoning Province: Shenyang City, on the campus of Shenyang Agricultural University, on the soil in meadows, 8 Sep 2019, X.D. Yu (SYAU-FUNGI-077); CHINA. Liaoning Province: Shenyang City, on the campus of Shenyang Agricultural University, on roadsides, 8 Sep 2019, H.B. Guo (SYAU-FUNGI-078).

**Remarks:** The main features of *M. subgriseoflava* are its greyish-brown to yellowish-grey pileus, creamy to light orange lamellae, greyish-yellow context, basidiospores with scattered warts and fusiform hymenial cystidia. On account of the pileus color, *M. subgriseoflava* is closely related to *M. griseoflava* originally described in northeastern China by *Pei et al. (2021)*. Nevertheless, *M. griseoflava*, differs from *M. subgriseoflava* by its adnate to adnexed and white lamellae. Besides, *M. griseoflava* differs by the presence of whitish tomentum at the stipe base (*Pei et al., 2021*). Micromorphologically, *M. griseoflava* is also distinct from *M. subgriseoflava* by its almost reticulate surface ornamentations of basidiospores and the presence of caulocystidia (*Pei et al., 2021*).

*Melanoleuca substridula* M. Zhang & X.D. Yu, sp. nov.

MycoBank No. MB 843804 (Fig. 4)

**Etymology:** The epithet "*substridula*" refers to the ochre brown color of the pileus, which is similar to the species *Melanoleuca stridula*.

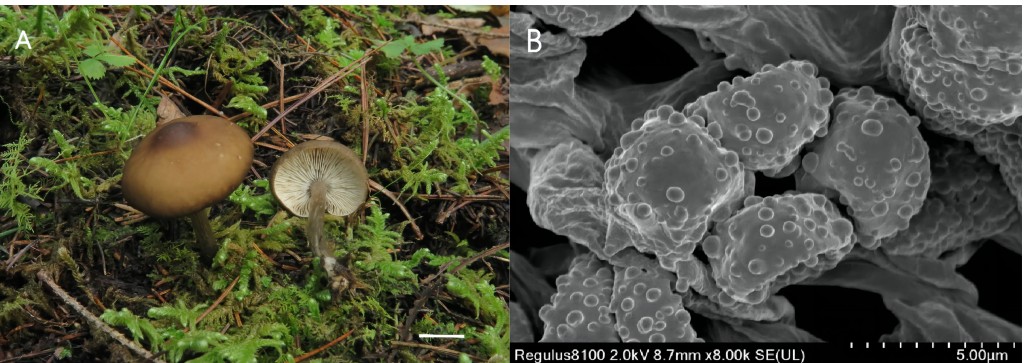

**Figure 4** *Melanouca substridula* (**Holotype, GDGM 84648**). (A) Macroscopic habitat; (B) surface of basidiospores. Scale bars: 1 cm (A) 5 μm (B).

**Diagnosis:** Basidiomata slightly small; pileus umbonate, brown to dark brown; lamellae sinuate to adnate, white; stipe light brown in upper part and grey-brown in lower part; basidiospores with round and scattered warts and lack of any forms of cystidia.

**Holotype:** CHINA. Sichuan Province: Jiuzhaigou valley reserve, on the soil in meadows, 19 Sep 2020, Ming Zhang (GDGM 84648).

**Description:** Basidiomata slightly small-sized. Pileus 21–38 mm diam.; umbonate at first; margin first slightly inflexed, soon becoming straight, depressed when mature and dry; surface glabrous, light brown at first (5D5 to 5D7), becoming brown to dark brown (7E8 to 7F8) when mature, often darker at margin, with a conspicuous dark brown (6F8 to 7F8) umbo at centre. Lamellae crowded, sinuate to adnate, 3.0–4.0 mm broad, white, edge entire and concolorous, with lamellulae of two or four lengths. Stipe cylindrical and somewhat broadened downwards, 26–38 mm long × 4.0-–5.0 mm diam., central, solid, light brown (6D4 to 6D8) in upper part, becoming grey-brown (5E3 to 6E3) towards base, with whitish flocculose apex, longitudinally striate, with whitish basal tomentum. Context up to 30–50 mm thick near stipe attachment, thinner at margin, white. Odor faint. Spore deposit whitish.

Basidiospores (86/6/2) 7.0–8.0 (9.0) × 5.0-–6.0 (6.5) μm, av. 7.4 × 5.2 μm, Q = (1.40) 1.42–1.45 (1.50), obovate to ellipsoid, subhyaline, ornamentation of somewhat regular warts, less warts with ridges, amyloid. Basidia (43/6/2) (25) 27–30 (31) × (6.0) 7.0–9.0 (10.0) μm, av. 30 × 8.0 μm, clavate, subhyaline. All types of cystidia absent. Lamella edge sterile. Trama hyphae thin-walled, regular, 5.5–16.5 μm wide. Pileipellis hyphae cylindrical, with numerous branched, thin-walled, up to 7.5 μm wide. Stipitipellis hyphae in parallel, with few branches, 3.5–8.0 μm wide, thin-walled. Caulocystidia absent. Clamp connections absent in all tissues.

**Habitat:** Solitary or in small groups, saprotrophic on the soil, on the grass in woods. Known from south-western China.

**Material studied:** CHINA. Sichuan Province: Jiuzhaigou valley reserve, 19 Sep 2020, Ming Zhang (GDGM 84683).

**Remarks:** The most distinctive characteristics of *Melanoleuca substridula* are its slightly small-sized basidiocarp, light brown to dark brown pileus with a prominent umbo, sinuate to adnate lamellae, light brown to greyish-brown stipe and lack of cystidia. On account of the pileus color, *M. substridula* is closely related to *M. stridula* originally described by *Singer (1943)*. However, *M. stridula* is featured by a slightly larger pileus (15–60 mm in diameter) (*Singer, 1943*). Additionally, *M. stridula* is often characterized by a pileus with a center depression, which differs by the umbonate pileus of *M. substridula*. Microscopically, *M. stridula* can be distinguishable from *M. substridula* by the presence of subcylindrical cystidia-like cells at the apex of the stipe (Metrod, 1949).

## DISCUSSION

Two new species, *Melanoleuca subgriseoflava* and *Melanoleuca substridula*, discovered and collected in Liaoning province and Sichuan province respectively, were originally reported and described in this study. Morphologically, the most distinctive features of *M. subgriseoflava* are a grey-brown to yellowish-grey pileus, creamy lamellae, fusiform hymenial cystidia and warted basidiospores. According to the classification system of *Boekhout (1988)*, *M. subgriseoflava* should belong to the section *Strictipedes* in the subgenus *Macrocystidia* because of the presence of grey-brown pileus and fusiform hymenial cystidia (*Boekhout, 1988*). Amongst the section *Strictipedes*, *M. subgriseoflava* mainly differs from the other species by its lack of caulocystidia, including *M. turrita* (Fr.) Sing, *M. polioleuca* (Fr.: Fr.) Kühn. & Maire, *M. atripes* Boekhout and *M. albifolia* Boekhout and (*Boekhout, 1988*). Furthermore, *M. albifolia* Boekhout differs from *M. subgriseoflava* by white lamellae. *M. polioleuca* (Fr.: Fr.) Kühn. & Maire can be distinguishable from *M. subgriseoflava* by a longer stipe, with a length of around 35–85 mm (*Boekhout, 1988*).

*Melanoleuca substridula* is easily recognized by its light brown to dark brown pileus, prominent umbo, adnate to sinuate lamellae, light brown to grey-brown stipe and acystidiate micromorphology. For lack of any forms of cystidia, *M. substridula* belongs to the subgenus *Melanoleuca* (*Boekhout, 1988*). Within the subgenus *Melanoleuca*, *M. graminicola* (Velen.) Kiihner& Maire, *M. melaleuca* (Pers.: Fr.) Murrill, *M. stridula* (Fr.) Metrod and *M. striimarginata* Metrod are characterized by adnate to subdecurrent lamellae and a depressed pileus center, making them easily distinguishable from *M. substridula* (*Boekhout, 1988*). The latter three species also differ by their larger basidiomata, with a pileus diameter of 35–65 mm. Moreover, *M. striimarginat a* differs by a striate margin of the pileus (*Metrod, 1942*).

Both the phylogenetic relationships and the divergence-time estimation, based on three regions (ITS-nrLSU-RPB2), showed that there are five clades in the genus *Melanoleuca* (Figs. 1 and 2), which was corroborated by *Pei et al. (2021)*. According to the phylogram, *M. subgriseoflava* is closely related to the other three species with high support in clade E, *i.e.*, *M. arcuata*, *M. heterocystidiosa* and *M. griseoflava*. *Melanoleuca arcuata* differs by its brick-red pileus and decurrent lamellae (*Fries, 1821*). *Melanoleuca heterocystidiosa* can be easily separated from *M. subgriseoflava* based on its smaller basidiomata, with a pileus diameter of 15 mm (Singer, 1939; *Bon, 1984*). *Melanoleuca griseoflava* differs from *M.*

*subgriseoflava* as elaborated above. In clade A, with the exception of *M. microcephala*, *M. substridula* is far away from the other species of *Melanoleuca*. However, *M. microcephala* can easily distinguish from *M. substridula* by its longer stipe with a length of 105 mm. Besides, caulocystidia in groups can be observed in *M. microcephala*, but not any forms of cystidia in *M. substridula* (*Antonín et al., 2021*).

In the present study, five clades can be recognized in both the BI tree (Fig. 1) and the MCC tree (Fig. 2). However, using phylogenetic analyses, species of *Melanoleuca* were divided into two clades in former research (*Vizzini et al., 2011*; *Yu et al., 2014*; *Nawaz, Jabeen & Khalid, 2017*; *Xu et al., 2019*). The species of *Melanoleuca* within each clade have lacked uniform characteristics to work in identification. In order to clarify the infrageneric classification, taxonomic treatments should be performed based on additional materials and complete morphological descriptions in later studies. Two new *Melanoleuca* species have been confirmed and a key for further studies on the *Melanoleuca* genus has been provided in this study.

### Funding
This study was supported by the National Natural Science Foundation of China (No. 31770014) and the Science and Technology Plan Project of Liaoning Province (2020-MZLH-33). The funders had no role in study design, data collection and analysis, decision to publish, or preparation of the manuscript.

### Grant Disclosures
The following grant information was disclosed by the authors:
National Natural Science Foundation of China: 31770014.
Science and Technology Plan Project of Liaoning Province: 2020-MZLH-33.

### Competing Interests
The authors declare there are no competing interests.

### Author Contributions
- Yue Qi performed the experiments, analyzed the data, prepared figures and/or tables, authored or reviewed drafts of the article, and approved the final draft.
- Cai-Hong Li performed the experiments, analyzed the data, authored or reviewed drafts of the article, and approved the final draft.
- Yu-Meng Song analyzed the data, authored or reviewed drafts of the article, and approved the final draft.
- Ming Zhang performed the experiments, prepared figures and/or tables, and approved the final draft.
- Hong-Bo Guo conceived and designed the experiments, prepared figures and/or tables, and approved the final draft.
- Xiao-Dan Yu conceived and designed the experiments, authored or reviewed drafts of the article, and approved the final draft.

## Data Availability

GenBank accession numbers ON262569 to ON262573 and ON220896 to ON220899.

## New Species Registration

The following information was supplied regarding the registration of a newly described species:

MycoBank No. MB843803

MycoBank No. MB 843804

## Supplemental Information

Supplemental information for this article can be found online at http://dx.doi.org/10.7717/peerj.13807#supplemental-information.

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
