# Peer review of "Melanoleuca subgriseoflava and M. substridula—two new Melanoleuca species (Agaricales, Basidiomycota) described from China"

_PeerJ, doi:10.7717/peerj.13807_

## Round 0.1 · original submission · Minor Revisions

Three experts revised the manuscript and found the content suitable for publication in Peer Journal. Please address the minor comments they provided.

·

Basic reporting

The work presented by Qi and colleagues shows that the two specimens of Melanolecua species collected in Chine are indeed two new species of this genus. The paper is written without flaws, is clear and contains enough information to support the findings.
The phylogenetic analysis is detailed and well designed and conducted. This unambiguously described the two specimens found as new species. The description of the specimens and the taxonomic analysis is sound and covers all the requirements for the description of new species.

Experimental design

The experimental design is sound and supports the findings reported in this manuscript. To this reviewer’s knowledge, the report is relevant to the field. The analysis conducted is thorough and supports the findings.

Validity of the findings

The details provided support the findings shown in this manuscript.

Additional comments

Overall, the manuscript is well written and contains all the details needed for the description of new species. I congratulate authors for this really good manuscript.
I kindly request that the Mycobank entries are available before publication, since the ID numbers provided cannot be accessed yet.
As minor comments:
In Line: I kindly request that authors provide a table with the primer sequence for each target to provide a manuscript that will be reference for other researchers, especially due to the large audience that Peer J has. Please, add this a supplementary material.

·

Basic reporting

no comment

Experimental design

no comment

Validity of the findings

no comment

Additional comments

This study reported two new Melanoleuca species, Melanoleuca subgriseoflava and M. substridula, collected in Liaoning and Sichuan provinces of China, respectively. The author used morphological and molecular methods to describe these two new species. The results showed that two new species derived from independent lineages. The English language were well written and the story was interesting. I advise this manuscript to be accepted after adding new necessary analyses.

Comments were listed below:
1. Divergence time of Melanoleuca species can be estimated to increase the depth of the manuscript.
2. Adding a section for estimation of evolutionary divergence among these Melanoleuca species sequences.
3. The parameter of SEM pictures need be removed and add a new bar beside every picture.

·

Basic reporting

The manuscript by Qi et al is a study of the identification of two new Melanoleuca species in China. The authors collected the materials in Liaoning and Sichuan provinces and conducted a careful morphological analysis. Phylogenetic analyses were also carried out to provide molecular evidence to validate the results. This work is well-executed. However, there are still some minor concerns below:
1. In the abstract, I suggest adding one or two sentence to emphasize the importance of the study. Similarly, the discussion part also should be polished up to make the reader understand the significance of this research.
2. There are some grammar mistakes in the MS. For example,Line 32, Line 75, Line 196, the correct article word should be added. Besides, Line 67 'remains scarce' should be 'remain scarce'. The authors should go through the whole MS carefully to correct grammar mistakes.
3. Some sentences are hard to read, such as Line 46-49. Some sentences or phrases are repeated many times. I suggest the authors may invite an English native professor to improve the language and help revise the MS.
4. I found the Author contributions and Conflicts of interest part are missing.

Experimental design

no comment

Validity of the findings

no comment

---

## Round 0.2 · accepted · Accept

The manuscript was improved after addressing the Reviewers' comments and it is now suitable for publication in PeerJ.